# ASID: Active Exploration for System Identification in Robotic Manipulation

**Marius Memmel, Andrew Wagenmaker, Chuning Zhu, Dieter Fox, Abhishek Gupta**
Paul G. Allen School of Computer Science & Engineering
University of Washington
Seattle, WA 98195, USA
{memmelma,ajwagen,zchuning,fox,abhgupta}@cs.washington.edu

## Abstract

Model-free control strategies such as reinforcement learning have shown the ability to learn control strategies without requiring an accurate model or simulator of the world. While this is appealing due to the lack of modeling requirements, such methods can be sample inefficient, making them impractical in many real-world domains. On the other hand, model-based control techniques leveraging accurate simulators can circumvent these challenges and use a large amount of cheap simulation data to learn controllers that can effectively transfer to the real world. The challenge with such model-based techniques is the requirement for an extremely accurate simulation, requiring both the specification of appropriate simulation assets and physical parameters. This requires considerable human effort to design for every environment being considered. In this work, we propose a learning system that can leverage a small amount of *real-world* data to autonomously refine a simulation model and then plan an accurate control strategy that can be deployed in the real world. Our approach critically relies on utilizing an initial (possibly inaccurate) simulator to design effective exploration policies that, when deployed in the real world, collect high-quality data. We demonstrate the efficacy of this paradigm in identifying articulation, mass, and other physical parameters in several challenging robotic manipulation tasks, and illustrate that only a small amount of real-world data can allow for effective sim-to-real transfer. Project website at https://weirdlabuw.github.io/asid

## 1 Introduction

Controlling robots to perform dynamic, goal-directed behavior in the real world is challenging. Reinforcement Learning (RL) has emerged as a promising technique to learn such behaviors without requiring known models of the environment, instead relying on data sampled directly from the environment (Schulman et al., 2017a; Haarnoja et al., 2018). In principle, these techniques can be deployed in new environments with a minimal amount of human effort, and allow for continual improvement of behavior. Such techniques have been shown to successfully learn complex behaviors in a variety of scenarios, ranging from table-top manipulation (Yu et al., 2020) to locomotion (Hwangbo et al., 2019) and even dexterous manipulation (Zhu et al., 2019).

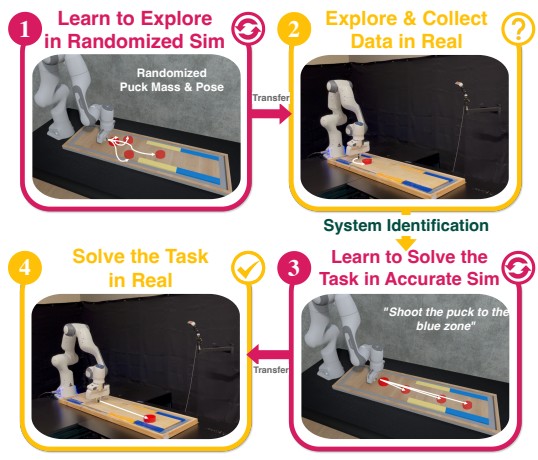

Figure 1: **ASID**: A depiction of our proposed process of active exploration for system identification, from learning exploration policies to real-world deployment.

However, these capabilities come at a cost, requiring full access to the environment in order to design reset mechanisms and reward functions. Such requirements necessitate training these methods in carefully controlled and often curated environments, limiting their applicability. Moreover, de-

ploying tabula-rasa reinforcement learning methods in the real world often requires a prohibitively large number of samples, which may be impractical to collect.

One approach to circumvent these challenges is to rely on *simulators* to cheaply generate large amounts of data and use RL to train a policy. However, directly deploying policies trained in simulation in the real world is often ineffective due to the discrepancy between the simulation and the real world, the so-called *sim2real gap*. For example, the physical properties and parameters of the simulation might diverge from the real world, rendering a simulation-trained policy useless in reality.

Taking inspiration from system identification, we argue that the key to effective sim2real transfer is an initial round of *exploration* in the real world to learn an effective simulator. We propose a generic pipeline for sim2real transfer, Active Exploration for System IDentification (ASID), which decouples exploration and exploitation: (1) exploration in real to collect informative data of unknown parameters, (2) refinement of the simulation parameters using data collected in real, (3) policy training on the updated simulator to accomplish the goal tasks. Our exploration procedure is motivated by work in theoretical statistics and seeks to induce trajectories corresponding to large *Fisher information*, thereby providing maximally informative observations. By using our initial round of exploration to obtain accurate estimates of the parameters in the real world, we show that in many cases, the policies trained in step (3) successfully transfer to real in a zero-shot fashion, even in settings where training a policy in sim without additional knowledge of real would fail.

A key insight in our approach is that, while a policy trained in sim to accomplish the goal task may not effectively transfer, strategies that explore effectively in sim often also explore effectively in real. As an example, say our goal task is to hit a ball to a particular location with a robotic arm and assume the mass of the ball is unknown. If we train a policy in sim to hit the ball without knowledge of the mass, when deployed in real it will almost certainly fail, as the force at which it should strike the ball depends critically on the (unknown) mass. To learn the mass, however, essentially any contact between the ball and the arm suffices. Achieving some contact between the ball and the arm requires a significantly less precise motion, and indeed, does not require any prior knowledge of the mass. We can therefore train a policy in sim that learns to effectively explore—hit the ball in any direction—and deploy this in real to collect information on the true parameters, ultimately allowing us to obtain a higher-fidelity simulator that *does* allow sim2real transfer on the goal task.

We are particularly interested in the application of our pipeline to modern robotic settings and evaluate ASID on four tasks: sphere manipulation, laptop articulation, rod balancing, and shuffleboard. We show that in all settings, our approach is able to effectively identify unknown parameters of the real environment (e.g. geometry, articulation, center of mass, and physical parameters like mass, friction, or stiffness), and using this knowledge, learn a policy in sim for the goal task that successfully transfers to real. In all cases, by deploying effective exploration policies trained in simulation, we require only a very small amount of data from real—typically a single episode of data suffices.

## 2 RELATED WORK

**System Identification:** Our work is closely related to the field of system identification (Åström & Eykhoff, 1971; Söderström & Stoica, 1989; Ljung, 1998; Schön et al., 2011; Menda et al., 2020), which studies how to learn a model of the system dynamics efficiently. A large body of work, stretching back decades, has explored how inputs to a system should be chosen to most effectively learn the system's parameters (Mehra, 1974; 1976; Goodwin & Payne, 1977; Hjalmarsson et al., 1996; Lindqvist & Hjalmarsson, 2001; Gerencsér & Hjalmarsson, 2005; Rojas et al., 2007; Gevers et al., 2009; Gerencsér et al., 2009; Manchester, 2010; Rojas et al., 2011; Bombois et al., 2011; Hägg et al., 2013; Wagenmaker & Jamieson, 2020; Wagenmaker et al., 2021; Mania et al., 2022; Wagenmaker et al., 2023) or how to deal with partial observability (Schön et al., 2011; Menda et al., 2020). Similar to our exploration strategy, many of these works choose their inputs to maximize some function of the Fisher information matrix. A primary novelty of our approach is to use a simulator to learn effective exploration policies, and to apply our method to modern, real-world robotics tasks—indeed, our work can be seen as bridging the gap between classical work on system identification and modern sim2real techniques. While the aforementioned works are primarily theoretical, recent work has studied the application of such methods to a variety of real-world settings like active identification of physics parameters (Xu et al., 2019; Kumar et al., 2019; Mavrakis et al., 2020; Gao et al., 2020; 2022) or kinematic structure (Mo et al., 2021; Wang et al., 2022; Nie et al., 2022; Hsu

et al., 2023) through object-centric primitives. Another line of recent work aims to learn the parameters of the simulator to ultimately train a downstream policy on the learned parameters, and therefore apply task-specific policies for data collection (Zhu et al., 2018; Chebotar et al., 2019; Huang et al., 2023; Ren et al., 2023) or exploration policies that minimize its regret (Liang et al., 2020). The majority of these works, however, do not consider running an exploration policy that targets learning the unknown parameters, do not address solving downstream tasks, or rely on techniques that do not scale effectively to more complex tasks.

**Simulation-to-Reality Transfer:** Transferring learned policies from *sim2real* has shown to be successful in challenging tasks like dexterous manipulation (OpenAI et al., 2018; Handa et al., 2022; Chen et al., 2022), locomotion (Rudin et al., 2022), agile drone flight (Sadeghi & Levine, 2016) or contact rich assembly tasks Tang et al. (2023), yet challenges remain due to the sim2real gap. To deal with the gap, Domain Randomization (DR) (Tobin et al., 2017) trains policies over a distribution of environments in simulation, hoping for the real world to be represented among them. Subsequent works adaptively change the environment distribution (Muratore et al., 2019; Mehta et al., 2020) and incorporate real data (Chebotar et al., 2019; Ramos et al., 2019; Duan et al., 2023; Chen et al., 2023; Ma et al., 2023; Torne et al., 2024). While similar to our approach, these methods do not perform targeted exploration in real to update the simulator parameters. Other approaches seek to infer and adapt to simulation parameters during deployment (Kumar et al., 2021; Qi et al., 2023; Margolis et al., 2023), leverage offline data (Richards et al., 2021; Bose et al., 2024), or adapt online (Sinha et al., 2022); in contrast, we do not learn such an online adaptation strategy, but rather a better simulator. Finally, a commonly applied strategy is to train a policy in sim and then fine-tune in the real environment (Julian et al., 2021; Smith et al., 2022; Nakamoto et al., 2023); in contrast, we are interested in the (more challenging) regime where a direct transfer is not likely to give any learning signal to fine-tune from.

**Model-Based RL:** Model-based RL (MBRL) aims to solve the RL problem by learning a model of the dynamics, and using this model to either plan or solve a policy optimization problem (Deisenroth & Rasmussen, 2011; Williams et al., 2017; Nagabandi et al., 2018; Chua et al., 2018; Janner et al., 2019; Hafner et al., 2019; 2020; Janner et al., 2022; Zhu et al., 2023). While our approach is model-based in some sense, the majority of work in MBRL focuses on fully learned dynamic models; in contrast, our "model" is our simulator, and we aim to learn only a very small number of parameters, which can be much more sample-efficient. Furthermore, MBRL methods typically do not perform explicit exploration, while a key piece of our approach is a targeted exploration procedure. The MBRL works we are aware of which do rely on targeted exploration (Shyam et al., 2019; Pathak et al., 2019) typically rely on fully learned dynamic models and apply somewhat different exploration strategies, which we show in Appendix A.3.2 can perform significantly worse.

## 3 PRELIMINARIES

We formulate our decision-making setting as Markov Decision Processes (MDPs). An MDP is defined as a tuple $M^\star = (\mathcal{S}, \mathcal{A}, \{P_h^\star\}_{h=1}^H, P_0, \{r_h\}_{h=1}^H)$, where $\mathcal{S}$ is the set of states, $\mathcal{A}$ the set of actions, $P_h : \mathcal{S} \times \mathcal{A} \to \triangle_{\mathcal{S}}$ the transition kernel, $P_0 \in \triangle_{\mathcal{S}}$ the initial state distribution, and $r_h : \mathcal{S} \times \mathcal{A} \to \mathbb{R}$ the reward function. We consider the episodic setting. At the beginning of an episode, the environment samples a state $s_1 \sim P_0$. The agent observes this state, plays some action $a_1 \in \mathcal{A}$, and transitions to state $s_2 \sim P_1(\cdot \mid s_1, a_1)$, receiving reward $r_1(s_1, a_1)$. After $H$ steps, the environment resets and the process repeats. Our primary goal is to learn a *policy* $\pi$—a mapping from states to actions—that maximizes reward in the true environment. We denote the value of a policy by $V_0^\pi := \mathbb{E}_{M^\star, \pi}[\sum_{h=1}^H r_h(s_h, a_h)]$, where the expectation is over trajectories induced playing policy $\pi$ on MDP $M^\star$. We think of the reward $r$ as encoding our *downstream task*, and our end goal is to find a policy that solves our task, maximizing $V_0^\pi$. We denote such policies as $\pi_{\text{task}}$.

In the sim2real setting considered in this work, we assume that the reward is known, but that the dynamics of the real environment, $P^\star = \{P_h^\star\}_{h=1}^H$, are initially unknown. However, we assume that they belong to some known parametric family $\mathcal{P} := \{P_{\boldsymbol{\theta}} : \boldsymbol{\theta} \in \Theta\}$, so that there exists some $\boldsymbol{\theta}^\star \in \Theta$ such that $P^\star = P_{\boldsymbol{\theta}^\star}$. Here we take $\boldsymbol{\theta}$ to be some unknown parameter (for example, mass, friction, etc.), and $P_{\boldsymbol{\theta}}$ the dynamics under parameter $\boldsymbol{\theta}$ (which we might know from physics, first principles, etc.). For any $\boldsymbol{\theta}$ and policy $\pi$, the dynamics $P_{\boldsymbol{\theta}}$ induce a distribution over state-action trajectories, $\boldsymbol{\tau} = (s_1, a_1, s_2, \ldots, s_H, a_H)$, which we denote by $p_{\boldsymbol{\theta}}(\cdot \mid \pi)$. We can think of

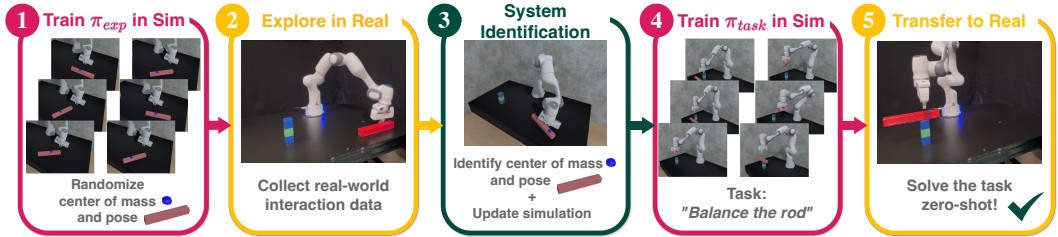

Figure 2: **Overview of ASID:** (1) Train an exploration policy $\pi_{\exp}$ that maximizes the Fisher information, leveraging the vast amount of cheap simulation data. (2) Roll out $\pi_{\exp}$ in real to collect informative data that can be used to (3) run system identification to identify physics parameters and reconstruct, *e.g.*, geometric, collision, and kinematic properties. (4) Train a task-specific policy $\pi_{\text{task}}$ in the updated simulator and (5) zero-shot transfer $\pi_{\text{task}}$ to the real world.

our simulator as instantiating $p_{\boldsymbol{\theta}}(\cdot \mid \pi)$—we assume our simulator is able to accurately mimic the dynamics of an MDP with parameter $\boldsymbol{\theta}$ under policy $\pi$, generating samples $\boldsymbol{\tau} \sim p_{\boldsymbol{\theta}}(\cdot \mid \pi)$.

In addition, we also assume that samples from our simulator are effectively "free"—for any $\boldsymbol{\theta}$ and policy $\pi$, we can generate as many trajectories $\boldsymbol{\tau} \sim p_{\boldsymbol{\theta}}(\cdot \mid \pi)$ as we wish. Given this, it is possible to find the optimal policy under $\boldsymbol{\theta}$ by simply running any standard RL algorithm in simulation. With knowledge of the true parameters $\boldsymbol{\theta}^{\star}$, we can then easily find the optimal policy in real by sampling trajectories from the simulated environment with parameter $\boldsymbol{\theta}^{\star}$. It follows that, if we can identify the true parameter $\boldsymbol{\theta}^{\star}$ in real, we can solve the goal task.

We consider the following learning protocol:

1. Learner chooses exploration policy $\pi_{\exp}$ and plays it in real for a *single* episode, generating trajectory $\boldsymbol{\tau}_{\text{real}} \sim p_{\boldsymbol{\theta}^{\star}}(\cdot \mid \pi_{\exp})$.
2. Using $\boldsymbol{\tau}_{\text{real}}$ and the simulator in any way they wish, the learner obtains some policy $\pi_{\text{task}}$.
3. Learner deploys $\pi_{\text{task}}$ in real and suffers loss $\max_{\pi} V_0^{\pi} - V_0^{\pi_{\text{task}}}$.

The goal of the learner is then to learn as much useful information as possible about the real environment from a single episode of interaction and use this information to obtain a policy that can solve the task in real as effectively as possible.

**Parameter Estimation and Fisher Information:** The *Fisher information matrix* plays a key role in the choice of our exploration policy, $\pi_{\exp}$. Recall that, for a distribution $p_{\boldsymbol{\theta}}$, satisfying certain regularity conditions, the Fisher information matrix is defined as:

$$\mathcal{I}(\boldsymbol{\theta}) := \mathbb{E}_{\boldsymbol{\tau} \sim p_{\boldsymbol{\theta}}} \left[ \nabla_{\boldsymbol{\theta}} \log p_{\boldsymbol{\theta}}(\boldsymbol{\tau}) \cdot \nabla_{\boldsymbol{\theta}} \log p_{\boldsymbol{\theta}}(\boldsymbol{\tau})^{\top} \right].$$

Assume that we have access to data $\mathfrak{D} = (\boldsymbol{\tau}_t)_{t=1}^{T}$, where $\boldsymbol{\tau}_t \sim p_{\boldsymbol{\theta}^{\star}}$ for $t = 1, \ldots, T$, and let $\widehat{\boldsymbol{\theta}}(\mathfrak{D})$ denote some unbiased estimator of $\boldsymbol{\theta}^{\star}$. Then the Cramer-Rao lower bound (see e.g. Pronzato & Pázman (2013)) states that, under certain regularity conditions, the covariance of $\widehat{\boldsymbol{\theta}}(\mathfrak{D})$ satisfies:

$$\mathbb{E}_{\mathfrak{D} \sim p_{\boldsymbol{\theta}^{\star}}}[(\widehat{\boldsymbol{\theta}}(\mathfrak{D}) - \boldsymbol{\theta}^{\star})(\widehat{\boldsymbol{\theta}}(\mathfrak{D}) - \boldsymbol{\theta}^{\star})^{\top}] \succeq T^{-1} \cdot \mathcal{I}(\boldsymbol{\theta}^{\star})^{-1}.$$

From this it follows that the Fisher information serves as a lower bound on the mean-squared error:

$$\mathbb{E}_{\mathfrak{D} \sim p_{\boldsymbol{\theta}^{\star}}}[\|\widehat{\boldsymbol{\theta}}(\mathfrak{D}) - \boldsymbol{\theta}^{\star}\|_2^2] = \text{tr}(\mathbb{E}_{\mathfrak{D} \sim p_{\boldsymbol{\theta}^{\star}}}[(\widehat{\boldsymbol{\theta}}(\mathfrak{D}) - \boldsymbol{\theta}^{\star})(\widehat{\boldsymbol{\theta}}(\mathfrak{D}) - \boldsymbol{\theta}^{\star})^{\top}]) \geq T^{-1} \cdot \text{tr}(\mathcal{I}(\boldsymbol{\theta}^{\star})^{-1}). \quad (1)$$

This is in general tight—for example, the maximum likelihood estimator satisfies (1) with equality as $T \to \infty$ (Van der Vaart, 2000). The Fisher information thus serves as a fundamental lower bound on parameter estimation error, a key motivation for our exploration procedure.

## 4 ASID: TARGETED EXPLORATION FOR TEST-TIME SIMULATION CONSTRUCTION, IDENTIFICATION, AND POLICY OPTIMIZATION

In this section, we present our proposed approach, ASID, a three-stage pipeline illustrated in Figure 2. We describe each component of ASID in the following.

## 4.1 Exploration via Fisher Information Maximization

As motivated in Section 3, to learn a policy effectively accomplishing our task, it suffices to accurately identify $\boldsymbol{\theta}^\star$. In the exploration phase, step 1 in our learning protocol, our goal is to then play an exploration policy $\pi_{\exp}$ which generates a trajectory on the real environment that provides as much information on $\boldsymbol{\theta}^\star$ as possible. Following Section 3, the Fisher information gives a quantification of the usefulness of the data collected, which motivates our approach.

In our setting, the distribution over trajectories generated during exploration in real, $\boldsymbol{\tau}_{\mathrm{real}} \sim p_{\boldsymbol{\theta}^\star}(\cdot \mid \pi_{\exp})$, depends on the exploration policy, $\pi_{\exp}$, being played. As the Fisher information depends on the data distribution, it too scales with the choice of exploration policy:

$$\mathcal{I}(\boldsymbol{\theta}^\star, \pi_{\exp}) := \mathbb{E}_{\boldsymbol{\tau} \sim p_{\boldsymbol{\theta}^\star}(\cdot \mid \pi_{\exp})} \left[ \nabla_{\boldsymbol{\theta}} \log p_{\boldsymbol{\theta}^\star}(\boldsymbol{\tau} \mid \pi_{\exp}) \cdot \nabla_{\boldsymbol{\theta}} \log p_{\boldsymbol{\theta}^\star}(\boldsymbol{\tau} \mid \pi_{\exp})^\top \right].$$

Following (1), if we collect trajectories by playing $\pi_{\exp}$ and set $\widehat{\boldsymbol{\theta}}$ to any unbiased estimator of $\boldsymbol{\theta}^\star$ on these trajectories, the mean-squared error of $\widehat{\boldsymbol{\theta}}$ will be lower bounded by $\mathrm{tr}(\mathcal{I}(\boldsymbol{\theta}^\star, \pi_{\exp})^{-1})$. The optimal exploration policy—the exploration policy which allows for the smallest estimation error—is, therefore, the policy which solves[1]

$$\arg\min_\pi \mathrm{tr}(\mathcal{I}(\boldsymbol{\theta}^\star, \pi)^{-1}). \tag{2}$$

As an intuitive justification for this choice of exploration policy, note that the Fisher information is defined in terms of the gradient of the log-likelihood with respect to the unknown parameter. Thus, if playing some $\pi_{\exp}$ makes the Fisher information "large", making $\mathrm{tr}(\mathcal{I}(\boldsymbol{\theta}^\star, \pi_{\exp})^{-1})$ small, this suggests $\pi_{\exp}$ induces trajectories that are very sensitive to the unknown parameters, *i.e.*, trajectory that are significantly more likely under one set of parameters than another. By exploring to maximize the Fisher information, we, therefore, will collect trajectories that are maximally informative about the unknown parameters, since we will observe trajectories much more likely under one set of parameters than another. Motivated by this, we therefore seek to play a policy during exploration that solves (2).

**Implementing Fisher Information Maximization:** In practice, several issues arise in solving (2), which we address here. First, the form of $\mathcal{I}(\boldsymbol{\theta}, \pi)$ can be quite complicated, depending on the structure of $p_{\boldsymbol{\theta}}(\cdot \mid \pi)$, and it may not be possible to efficiently obtain a solution to (2). To address this, we make a simplifying assumption on the dynamics, that our next state, $s_{h+1}$, evolves as:

$$s_{h+1} = f_{\boldsymbol{\theta}}(s_h, a_h) + w_h, \tag{3}$$

where $s_h$ and $a_h$ are the current state and action, $w_h \sim \mathcal{N}(0, \sigma_w^2 \cdot I)$ is Gaussian process noise, and $f_{\boldsymbol{\theta}}$ are the nominal dynamics. Under these dynamics, the Fisher information matrix reduces to

$$\mathcal{I}(\boldsymbol{\theta}, \pi) = \sigma_w^{-2} \cdot \mathbb{E}_{p_{\boldsymbol{\theta}}(\cdot \mid \pi)} \left[ \sum_{h=1}^H \nabla_{\boldsymbol{\theta}} f_{\boldsymbol{\theta}}(s_h, a_h) \cdot \nabla_{\boldsymbol{\theta}} f_{\boldsymbol{\theta}}(s_h, a_h)^\top \right].$$

We argue that solving (2) with this form of $\mathcal{I}(\boldsymbol{\theta}, \pi)$ is a very intuitive objective, even in cases when the dynamics may not follow (3) exactly. Indeed, this suggests that during exploration, we should aim to reach states for which the dynamics $f_{\boldsymbol{\theta}}$ have a large gradient with respect to $\boldsymbol{\theta}$—states for which the next state predicted by the dynamics is very sensitive to $\boldsymbol{\theta}$. In such states, observing the next state gives us a significant amount of information on $\boldsymbol{\theta}$, allowing us to accurately identify $\boldsymbol{\theta}^\star$.

A second challenge in solving (2) is that we do not know the true parameter $\boldsymbol{\theta}^\star$, which the optimization (2) depends on. To circumvent this, we rely on domain randomization in choosing our exploration policy, solving instead:

$$\pi_{\exp} = \arg\min_\pi \mathbb{E}_{\boldsymbol{\theta} \sim q_0} [\mathrm{tr}(\mathcal{I}(\boldsymbol{\theta}, \pi)^{-1})] \tag{4}$$

for some distribution over parameters $q_0$. While this is only an approximation of (2), in practice we find that this approximation yields effective exploration policies since, as described in Section 1, in many cases exploration policies require only a coarse model of the dynamics, and can therefore often be learned without precise knowledge of the unknown parameters.

A final challenge is that, in general, we may not have access to a differentiable simulator, and our dynamics themselves may not be differentiable. In such cases, $\nabla_{\boldsymbol{\theta}} f_{\boldsymbol{\theta}}(s_h, a_h)$ is unknown or

---

[1]In the experiment design literature, this is known as an *A-optimal experiment design* (Pukelsheim, 2006).

undefined, and the above approach cannot be applied. As a simple solution to this, we rely on a finite-differences approximation to the gradient, which still provides an effective measure of how sensitive the next state is to the unknown parameter. In practice, to solve (4) and obtain an exploration policy, we rely on standard policy optimization algorithms, such as PPO (Schulman et al., 2017b).

## 4.2 System Identification

ASID runs the exploration policy $\pi_{\mathrm{exp}}$ (Section 4.1) in the real environment to generate a single trajectory $\boldsymbol{\tau}_{\mathrm{real}} \sim p_{\boldsymbol{\theta}^{\star}}(\cdot \mid \pi_{\mathrm{exp}})$. In the system identification phase, ASID then updates the simulator parameters using the collected trajectory. The goal is to find a distribution over simulator parameters that yield trajectories that match $\boldsymbol{\tau}_{\mathrm{real}}$ as closely as possible. In particular, we wish to find some distribution over simulation parameters, $q_{\boldsymbol{\phi}}$, which minimizes:

$$\mathbb{E}_{\boldsymbol{\theta} \sim q_{\boldsymbol{\phi}}}[\mathbb{E}_{\boldsymbol{\tau}_{\mathrm{sim}} \sim p_{\boldsymbol{\theta}}(\cdot \mid \mathcal{A}(\boldsymbol{\tau}_{\mathrm{real}}))}[\|\boldsymbol{\tau}_{\mathrm{real}} - \boldsymbol{\tau}_{\mathrm{sim}}\|_2^2]]$$

where $p_{\boldsymbol{\theta}}(\cdot \mid \mathcal{A}(\boldsymbol{\tau}_{\mathrm{real}}))$ denotes the distribution over trajectories generated by the simulator with parameter $\boldsymbol{\theta}$, and playing the same sequence of actions as were played in $\boldsymbol{\tau}_{\mathrm{real}}$. In practice, we apply REPS (Peters et al., 2010) for the simulation and Cross Entropy Method (CEM) for the real-world experiments. We stress that the ASID framework is generic, and other black-box optimization algorithms could be used instead.

## 4.3 Solving the Downstream Task

After exploration and identification, the simulator can include information about the kinematic tree, position, orientation, or size of the object of interest, and the physical parameters of the real environment. With such a high-fidelity simulator, we aim to solve the downstream tasks entirely in simulation and transfer the learned policies $\pi_{\mathrm{task}}$ to the real world in a zero-shot fashion. As with the system identification stage, ASID does not assume a particular method for solving the downstream task and any policy optimization algorithm can be used.

## 5 Experimental Evaluation

In our experimental evaluation, we aim to answer the following questions:

1. Does ASID's exploration strategy yield sufficient information to identify unknown parameters?

2. Are downstream task policies learned via ASID successful when using the identified parameters?

3. Does the paradigm suggested by ASID transfer to performing tasks in the real world?

We conduct our experimental evaluation in two scenarios. First, we conduct empirical experiments entirely in simulation (Todorov et al., 2012) to validate the behavior of the exploration, system identification, and downstream task modules of ASID. This involves treating a true simulation environment as the real-world environment and then a reconstruction of this simulation environment as the approximate constructed simulation. Policies learned in this approximate simulation can then be evaluated back in the original simulation environment to judge efficacy. Second, we apply this to two real-world manipulation tasks that are inherently dependent on accurate parameter identification, showing the ability of ASID to function in the real world, using real-world data.

### 5.1 Ablations and Baseline Comparisons

We compare ASID with several baselines and ablations for various portions of the pipeline.

**Exploration:** To understand the importance of targeted exploration via Fisher information maximization we compare with two baselines. First, we compare with the naïve approach of using data from a random policy for system identification that is not performing any targeted exploration. Secondly, we compare with Kumar et al. (2019), which aims to generate exploration trajectories that maximize mutual information with the parameters of interest. This method essentially learns a parameter inference network and then rewards exploration policies for minimizing its error.

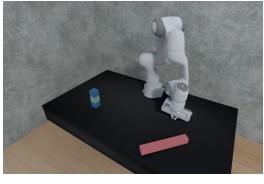 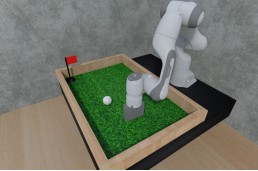 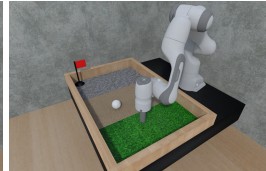 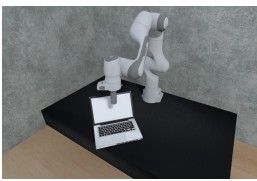

|  (a) Rod balancing | (b) Sphere manipulation | (c) Multi-friction | (d) Articulation |

Figure 3: **Depiction of environments in simulation**

**System Identification:** To evaluate the impact of system identification methods in this pipeline, we compare the effectiveness of using optimization-based system identification, e.g., (Peters et al., 2010; Memmel et al., 2022), with an end-to-end learned module (Kumar et al., 2019). This comparison shows the stability and extrapolation benefits of ASID over completely data-driven modeling techniques. In particular, we evaluate ASID with the optimization-based system identification replaced by the learned estimation module from Kumar et al. (2019) (ASID + estimator).

**Downstream Policy Learning:** Since the eventual goal is to solve the downstream task, we finally compare how effective it is to learn downstream policies in simulation as opposed to an uninformed Domain Randomization (DR) over the parameters, in contrast to the targeted and identified parameters that stem from the ASID pipeline.

## 5.2 SIMULATED TASK DESCRIPTIONS

**Sphere Manipulation:** We consider two sphere manipulation tasks where physical parameters like rolling friction, object stiffness, and tangential surface friction are unknown and make a significant impact on policy performance: 1) striking a golf ball to reach a goal with unknown system friction (Figure 3b), 2) striking a golf ball in an environment with multiple patches that experience different surface friction (Figure 3c). In each scenario, the position of objects and the physical parameters are varied across evaluations. In this setting, we train downstream tasks with PPO.

**Rod Balancing:** Balancing, or dynamic stacking of objects critically depends on an accurate estimate of the inertia parameters. We construct a rod balancing task where the agent can interact with a rod object (Figure 3a) to identify its physical parameters, in this case varying the distribution of mass along the rod. Once the parameters are identified, the rod must be balanced by placing it on a ledge (Figure 5 (left)). The error is measured by the tilting angle of the rod after placement, a task that requires accurate estimation of system parameters. In this case, we use the CEM to optimize downstream policies.

**Articulation:** To stress generality, we consider environments that don't simply identify physical parameters like mass or friction but also the structure of the kinematic tree such as articulation and joint positioning. We consider an environment involving an articulated laptop model with a binary environment parameter representing whether articulation is present or not (Figure 3d).

## 5.3 DOES ASID LEARN EFFECTIVE EXPLORATION BEHAVIOR?

To evaluate the exploration behavior quantitatively, we consider the case of multiple unknown parameters, where learning each parameter requires exploration in a different region. In particular, we compare the exploration policy learned by our method with the exploration method of Kumar et al. (2019), on the multi-friction sphere manipulation environment illustrated in Figure 3c where the surface exhibits three different friction parameters (indicated by the grass, sand, and gravel textures). We initialize the golf ball in the leftmost region (grass)—to explore, the arm must strike the ball to other regions to identify their friction parameters. We plot a heat map of the regions visited by the ball during exploration in Figure 4. As can be seen, our approach achieves roughly uniform coverage over the entire space, learning to strike the ball into each region, and illustrating that our method is able to effectively handle complex exploration tasks that involve exciting multiple parameters. In contrast, the approach of Kumar et al. (2019) does not effectively move the sphere to regions different from the starting region.

Table 1: **Downstream task results in simulation:** Random exploration fails in tasks where directed exploration is required, *e.g.*, striking a sphere or finding the inertia of a rod. When placing the rod with a single action, domain randomization (DR) cannot solve the task without knowing the physical parameters. Learned system identification (Kumar et al. (2019) and ASID + estimator) doesn't generalize to unseen trajectories and becomes far less effective than optimization-based system identification (*cf.* ASID + SysID).

| Task | **Rod Balancing** | | | **Sphere Striking** |
|---|---|---|---|---|
| Metric | Tilt angle in degree° ↓ | | | Success Rate in % ↑ |
| Parameter | Inertia (left) | Inertia (middle) | Inertia (right) | Friction $\sim [1.1, 1.5]$ |
| Random exploration | $12.44 \pm 19.6$ | $4.20 \pm 6.5$ | $15.34 \pm 15.9$ | $10.62 \pm 4.3$ |
| Kumar et al. (2019) | $13.70 \pm 9.3$ | $2.82 \pm 2.7$ | $15.26 \pm 9.8$ | $9.50 \pm 2.4$ |
| DR | $26.69 \pm 7.0$ | $13.05 \pm 7.3$ | $1.13 \pm 1.3$ | $8.75 \pm 1.5$ |
| ASID + estimator | $17.73 \pm 13.1$ | $4.65 \pm 5.1$ | $9.99 \pm 6.8$ | $11.00 \pm 5.2$ |
| **ASID + SysID (ours)** | $\mathbf{0.00 \pm 0.0}$ | $\mathbf{0.72 \pm 1.0}$ | $\mathbf{0.00 \pm 0.0}$ | $\mathbf{28.00 \pm 9.7}$ |

We further analyze the exploration behavior in the articulation environment (Figure 3d). Here our exploration policy interacts with the laptop $80\%$ of the time as opposed to $20\%$ for naïve baselines. Appendix A.2.2 shows that models trained to predict articulation from interaction data, *e.g.*, Ditto (Jiang et al., 2022), can infer joint and part-level geometry from the collected data.

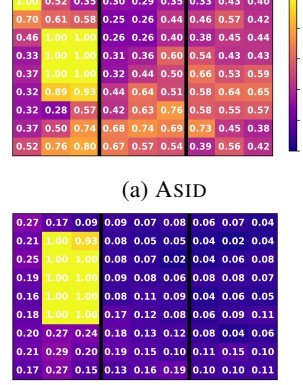

(a) ASID

(b) Kumar et al. (2019)

Figure 4: **Visitation frequency** of the sphere when explored by different exploration policies on multi-friction (Figure 3c). ASID activates the sphere over a much larger area, thereby identifying parameters more accurately

### 5.4 How does ASID perform quantitatively in simulation on downstream tasks?

To quantitatively assess the benefits of ASID over baselines and ablations, we evaluate the performance of ASID with respect to the baselines in simulation on downstream evaluation for the rod balancing (Figure 3a) and sphere manipulation (Figure 3c) task. Results are given in Table 1. Observing the performance on the rod environment, it becomes clear that previous methods struggle with one-shot tasks. When the inertia is not properly identified, CEM predicts the wrong action with high certainty, causing it to pick and place the rod at the wrong location. Domain randomization (DR) on the other hand tries to learn a policy over all possible mass distributions which in this case leads to a policy that always grasps the object at the center of the reference frame. The success here depends on "getting lucky" and sampling the correct action for the true inertia parameter.

In the sphere environment, the distance rolled and bounce of the sphere are critically dependent on parameters such as friction and stiffness, and misidentifying these parameters can lead to significantly different trajectories and unsuccessful downstream behavior. This is reflected in the significantly higher success rate of ASID as compared to baselines that are uninformed of parameters and simply optimize for robust policies (DR), those that try to use learned estimators for exploration (Kumar et al., 2019) or using random exploration.

### 5.5 Does ASID allow for real-world controller synthesis using minimal real-world data?

We further evaluate ASID on real-world tasks, replicating the rod balancing task from simulation (Figure 5) and introducing a novel shuffleboard task (Figure 6). As in simulation, we use a Franka Emika Panda robot for exploration and task performance in the real world. We compute the object's position and orientation by color-threshold the pointclouds from two calibrated Intel RealSense D455 cameras—this approach could easily be replaced by a more sophisticated pose estimation system if desired.

**Rod Balancing:** The goal of this task is to properly balance a rod with an unknown mass distribution by informatively exploring the environment in the real world and using the resulting data to identify the appropriate physics parameters in simulation. The optimized controller in simulation is then deployed to perform the task in the real world. In this case, the downstream task is picking the rod at a certain point along its axis and balancing it on a perch (Figure 1, Figure 5).

The policy executes the downstream task by a pick and place primitive parameterized by the exact pick point. We deploy ASID fully autonomously, executing exploration, system identification, downstream task training, and execution in an end-to-end fashion.

The mass distribution in the rod is varied and both the inertia and the environment friction must be inferred. While ASID correctly identifies the inertia of the rod most of the time, we find that a center of mass close to the middle of the rod causes ambiguities that hurt our system identification process causing the simulation to not be accurate enough for zero-shot transfer. Overall, ASID solves the rod-balancing task **6/9** times across varying mass distribution and pose while a domain-randomization policy trained without any environment interaction fails to solve it at all (Table 2).

**Shuffleboard:** This task is a scaled-down version of the popular bar game tabletop shuffle-

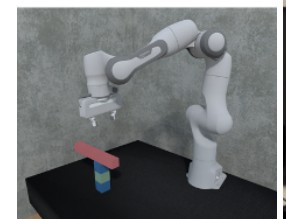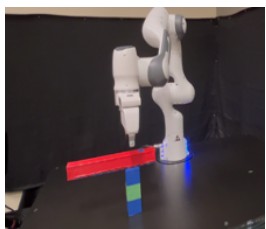

Figure 5: **Real-world Rod Balancing:** Simulation setup for training exploration and downstream task policies (left). Successful execution of autonomous real-world rod balancing with skewed mass (right).

Table 2: **Downstream task results in real:** ASID successfully balances the rod while domain randomization (DR) fails catastrophically.

| Task | **Rod Balancing** | | |
|---|---|---|---|
| Inertia | left | middle | right |
| DR | 0/3 | 0/3 | 0/3 |
| **ASID (ours)** | **2/3** | **1/3** | **3/3** |

board where a puck must be shot to a target area on a slippery board. We closely follow the original game and pour wax (sand) on the board to decrease the surface friction. This modification makes the task especially difficult as the surface friction on the board changes slightly after each shot since the puck displaces the wax. The goal is to strike the puck to one of the target regions—one closer (yellow) and one further (blue) away from the robot (Figure 2, Figure 6). After exploring the scene, ASID estimates the sliding and torsional friction of the puck to account for the changing surface friction of the board. For executing the downstream task, a primitive positions the endeffector at a fixed distance to the puck and a policy predicts a force value that parameterizes a shot attempt.

Due to the changing surface friction, the domain randomization baseline struggles to shoot the puck to the desired zone. While it succeeds 3/10 times—probably because the surface friction was part of its training distribution—the other attempts fall short or overshoot the target. With its dedicated exploration phase, ASID can accurately adapt the simulation to the current conditions and lands the puck in the desired zone **7/10** times (Table 3).

## 6 DISCUSSION

In this work, we introduced a novel pipeline for performing real-world control by autonomously exploring the environment, using the collected data for system identification, and the resulting identified system for downstream policy optimization. In essence, this sim-to-real-to-sim-to-real pipeline allows for targeted test-time construction of simulation environments in a way that enables the performance of downstream tasks. We demonstrate how this

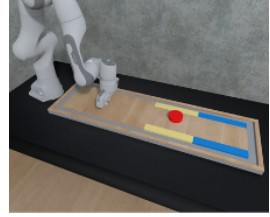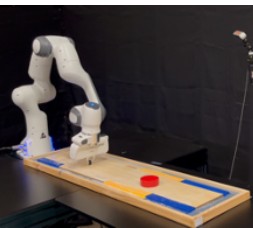

Figure 6: **Real-world Shuffleboard:** Simulation setup for training exploration and downstream task policies (left). Successful strike to the yellow zone (right).

Table 3: **Downstream task results in real:** ASID outperforms domain randomization (DR) on shooting the puck to the desired zones.

| Task | **Shuffleboard** | |
|---|---|---|
| Target zone | yellow (close) | blue (far) |
| DR | 2/5 | 1/5 |
| **ASID (ours)** | **4/5** | **3/5** |

type of identification requires autonomous and carefully directed exploration, and introduce a novel algorithm based on Fisher information maximization that is able to accomplish this directed exploration. The autonomously collected trajectories can then be paired with downstream optimization-based system identification and reconstruction algorithms for accurate simulation construction and downstream policy learning. We show the efficacy of this paradigm on multiple environments in simulation, as well as on rod balancing and shuffleboard, two challenging real-world tasks.

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

# A Appendix

## A.1 Task Details

### A.1.1 Sphere Manipulation

In the sphere manipulation tasks, the observation space consists of endeffector position, sphere position, and robot joint angles. During the training of the exploration policy, we randomize the location of a sphere with $r = 0.03$ to be between $x \in [0.430.65], y \in [-0.2, 0.23]$ for training and $x \in [0.55, 0.65], y \in [-0.2, 0.23]$ for evaluation. Parameter ranges for training are 1) friction $\theta \in [1e - 3, 5e - 3]$ and 2) patch friction $\theta_0, \theta_1, \theta_2 \in [1e - 5, 1e - 3]$ and set fixed sphere and goal locations as well as parameters for the downstream task. For 1) we attach a paddle and limit the endeffector position such that the robot has to strike the sphere to the goal.

### A.1.2 Articulation

In the case of the articulation environment, the parameter is binary and indicates whether articulation is present or not. During evaluation, articulation is always turned on and success is indicated by $\delta\beta > 1e - 2$. The initial laptop state is randomized over position $x \in [0.45, 0.65], y \in [-0.1, 0.1], yaw \in [0.00, 3.14]$ and lid angle $\beta \in [1.2, 2.5]$. The observation space contains the endeffector position, joint angles, position, orientation, and joint angle of the laptop.

### A.1.3 Rod Balancing

Similar to the sphere manipulation, we restrict the exploration policy to endeffector control with $\delta x, \delta y$ and attach a peg to the Franka. The rod has dimensions $0.04 \times 0.3 \times 0.04$ and initializes as $x \in [0.5, 0.6], y \in [-0.2, 0.2], yaw \in [0.00, 3.14]$

## A.2 QUALITATIVE RESULTS

### A.2.1 EXPLORATION STRATEGIES LEARNED BY THE AGENT

We evaluate the exploration behavior qualitatively across multiple environments to understand whether the exploration behavior is meaningful. In the sphere environment, we observe the agent hitting the sphere multiple times when it bounces off the walls or stays within reach. The other environments also experience emergent behavior, as the agent executes a horizontal motion towards the top of the lid if the laptop is mostly open while an almost closed laptop causes it to push from top to bottom instead. Finally, when determining the rod's inertia, the policy pushes both sides to gain maximum information about the center of mass through the rotation motions in both directions. See Figure 7 for a visualization of the exploration for the sphere environment.

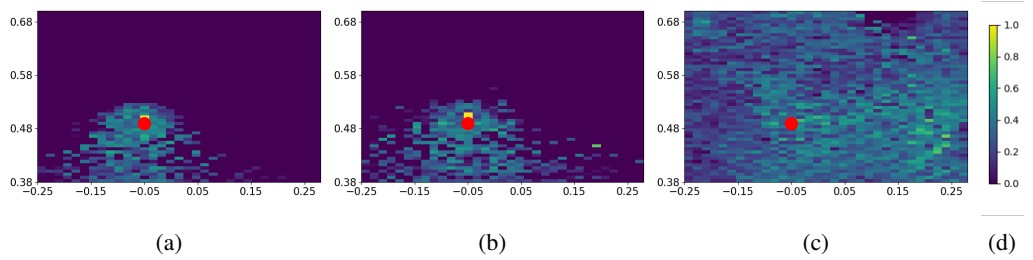

(a)        (b)        (c)    (d)

Figure 7: **Absolute sphere displacement** for different sphere starting locations. Zero means the sphere didn't get hit, higher numbers denote larger displacements. Initial endeffector position marked in red ■. a) Random Coverage, b) PPO Coverage, c) Fisher Coverage, d) Legend.

### A.2.2 RECONSTRUCTING ARTICULATION FROM INTERACTION DATA

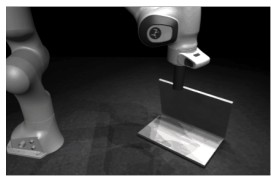  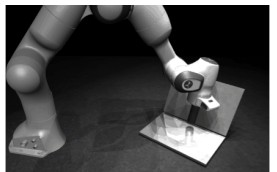  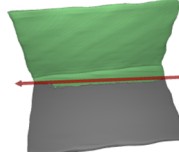

Figure 8: **Articulated object:** before (left) and after (middle) exploration with ASID and part-level reconstruction with Ditto (Jiang et al., 2022): articulated ■ and static part ■ (right).

## A.3 COMPARISON TO MODEL-BASED APPROACHES

### A.3.1 ENVIRONMENT SETUP

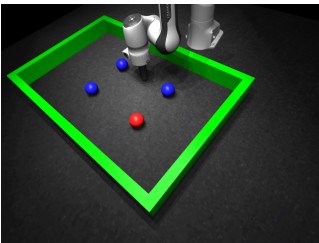

Figure 9: **Environment with multiple spheres.** The single red sphere ■ is subject to changing friction values while the three blue spheres ■ act as distractors.

### A.3.2  MODEL-BASED EXPLORATION

In this section, Figure 10, we compare the performance of our algorithm to that of the MAX algorithm (Shyam et al., 2019).

MAX aims to cover the state space through multiple policies throughout training. Since it uses the disagreement between fully learned dynamics models, it gets distracted by novel states induced by the movement of all spheres (red and blue). In contrast, our method based on the Fisher information yields a single policy that seeks out the sphere affected by the changing physics parameters (red) and ignores the irrelevant spheres (blue) even if they lead to novel states.

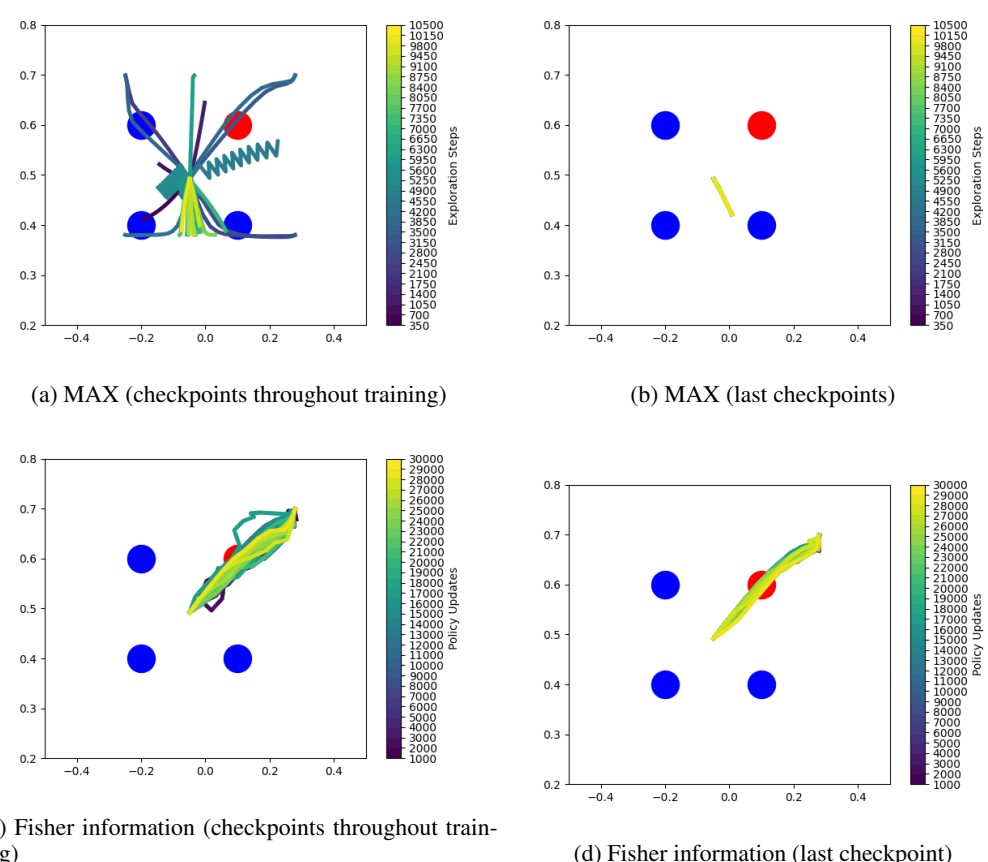

(a) MAX (checkpoints throughout training)

(b) MAX (last checkpoints)

(c) Fisher information (checkpoints throughout training)

(d) Fisher information (last checkpoint)

Figure 10: **Exploration MAX vs. Fisher information:** Trajectories collected (30) from policy checkpoints throughout training or from the last policy checkpoint. While Model-based Active Exploration (MAX) (Shyam et al., 2019) explores the state space over the course of training, getting distracted by the novel states induced by the blue spheres ■ (*c.f.* fig. 10a), Fisher information-based exploration (ours) shows directed exploration (*c.f.* fig. 10d), moving towards the sphere with changing parameters (red ■ )and yielding a single exploration policy.

### A.3.3   Learned state-transition models vs. simulators

We next illustrate the improvement using a simulator vs a fully learned dynamics model can give. We train a forward dynamics model (three layer MLP) on data generated both from MAX and our exploration procedure. When evaluated on out-of-distribution trajectories, i.e., trajectories not included in the training data, we find the model to be extremely inaccurate (see Figure 11). While the simulator extrapolates to unseen states and correctly predicts the movement of the sphere, the model hallucinates movement even when the end-effector does not interact with it at all! These findings make ASID preferable to a purely model-based approach.

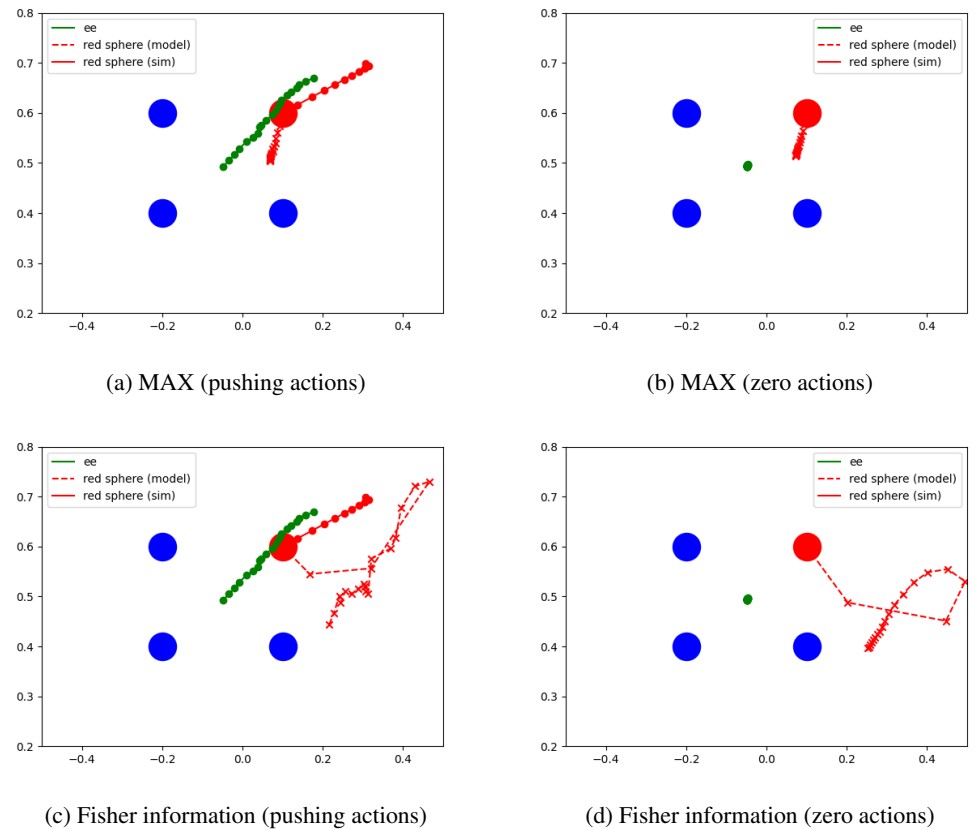

(a) MAX (pushing actions)

(b) MAX (zero actions)

(c) Fisher information (pushing actions)

(d) Fisher information (zero actions)

Figure 11: **Learned models vs. simulator:** Evaluation of a forward model trained on trajectories (30) from MAX (Shyam et al., 2019) (checkpoints throughout training) and Fisher information (last checkpoint). In contrast to our simulator, the learned model fails to predict the red sphere's ■ trajectory accurately even under no contact scenarios (*c.f.* fig. 11b, fig. 11d).

## A.4 POLICY GENERALIZATION

We evaluate the generalization capabilities of our exploration policy for different sphere locations and parameter combinations seen and unseen during training (Figure 13). Since we train the policies using an arena setup, we remove the arena during this evaluation to be able to query sphere locations outside of it (Figure 12).

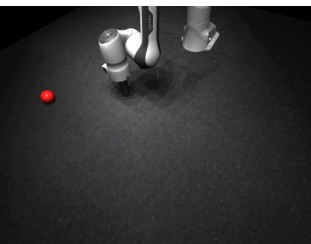

Figure 12: **Environment with free space.** The red sphere ■ is subject to changing parameter values and position is randomized.

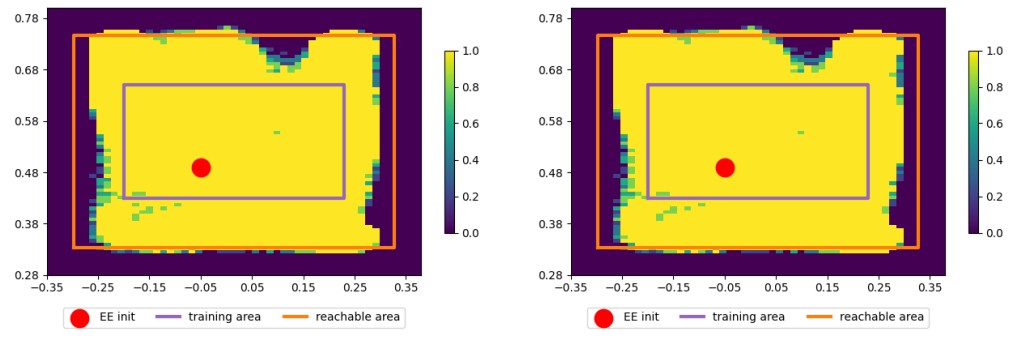

(a) Changing friction parameters (in distribution)  (b) Changing mass parameter (out of distribution)

Figure 13: **Generalization capabilities** of an exploration policy trained on sphere friction for different sphere starting locations. The plot shows the success rate of pushing the sphere spawned at the corresponding x-y-location and evaluated over 5 different seeds, i.e., different physics parameter values. Initial endeffector position marked in red ■. Boxes denote sphere locations seen during training (purple box ■) and reachable area of the endeffector (orange box ■). Our policies experience surprising generalization capabilities to unseen sphere locations as long as they can be reached by the robot (cf.fig. 13a). Furthermore, the actions extrapolate to unseen physics parameters as long as they can be uncovered with similar interactions, e.g., pushing (cf.fig. 13b). This is the case because the policy does not have any information about the underlying physics parameters until it hits the sphere.

