# OpenReview forum: "ASID: Active Exploration for System Identification in Robotic Manipulation"
_ICLR.cc/2024/Conference — ICLR 2024 oral_

### Official Review · Reviewer_4bkX · 2023-10-17

**Soundness:** 2 fair
**Presentation:** 2 fair
**Contribution:** 2 fair
**Rating:** 6
**Confidence:** 3

**Summary:**

This paper proposes a framework for model-based RL aimed at learning model parameters as well as the optimal policy given the model. The method seeks to address to the sim-to-real gap by proposing an efficient policy for exploring the environment inasmuch as that exploration improves the model. At each step, the method finds the policy that approximately maximizes the Fisher Information in the trajectories we expect the policy to encounter when rolled out. The method is experimentally validated on a number of real world environments.

**Strengths:**

- The paper addresses an important problem, i.e. directing exploration of an environment in an effort to reduce model uncertainty.
- The paper proposes a seemingly novel approach of finding policies that maximize the Fisher information.
- The paper validates the approach using real-world experiments

**Weaknesses:**

- Presentation/clarity can be improved: specifically, the abstract and introduction mostly describe the field of active exploration for system identification and adaptive control, as opposed to the specific method proposed, which appears to overstate the paper’s novelty. Furthermore, the approach warrants a better intuitive explanation. As I understand it, the Fisher Information objective attempts to quantify the sensitivity of model parameters to trajectories expected given some policy. Therefore, maximizing this objective yields a policy that, when executed, yields the maximum additional information about the model parameters.
- Lack of baselining/adequate discussion of other methods that use the Fisher information objective. The statement “As compared to these works, a primary novelty of our approach is the use of a simulator to learn effective exploration policies” seems too strong and overstated given that there are entire fields dedicated to this, and “the application of our method to modern, real-world robotics tasks” is an inadequate claim to novelty.
- Literature review can be improved with a discussion of the following:
    - Bayesian RL/Bayes-adaptive MDPs: 	M. Duff. Optimal Learning: Computational Procedure for Bayes-Adaptive Markov Decision Processes.  PhD thesis, University of Massachusetts, Amherst, USA, 2002.
    - PILCO:
        - Deisenroth, Marc, and Carl E. Rasmussen. "PILCO: A model-based and data-efficient approach to policy search." Proceedings of the 28th International Conference on Machine Learning (ICML-11). 2011.
    - Adaptive MPC:
        - S. M. Richards, N. Azizan, J.-J. Slotine, and M. Pavone. Adaptive-control-oriented meta-learning for nonlinear systems. In Robotics: Science and Systems, 2021. URL https://arxiv.org/abs/2204.06716.
        - Sinha, Rohan, et al. "Adaptive robust model predictive control with matched and unmatched uncertainty." 2022 American Control Conference (ACC). IEEE, 2022.
    - System identification in partially observable environments:
        - Menda, Kunal, et al. "Scalable identification of partially observed systems with certainty-equivalent EM." International Conference on Machine Learning. PMLR, 2020
        - Schön, Thomas B., Adrian Wills, and Brett Ninness. "System identification of nonlinear state-space models." Automatica 47.1 (2011): 39-49.

**Questions:**

1. In Section 4.2.1, I was expecting to see the standard SysID loss, which is to maximize the likelihood of the data (in this case trajectories) given model parameters. You find the distribution that maximizes likelihood for domain randomization. It seems to me that without some sort of entropy maximization term in the objective, or bootstrap, you would just end up with an MLE objective, whereas it seems like you want to find the Bayesian posterior of models given data. Can you comment on how your objective relates to that of finding a Bayesian posterior?
2. For a paper proposing active exploration for the sake of system identification, I wanted to see more discussion of the following: a) regret minimization, i.e. can you prove that your method minimizes regret and achieves the best policy with the fewest interactions with the environment? b) identifiability, i.e. can you say anything about whether all system parameters will be uniquely identified with infinite interactions?

---

> ### Author Response · Authors · 2023-11-18
>
> __”Presentation/clarity can be improved: specifically, the abstract and introduction mostly describe the field of active exploration for system identification and adaptive control, as opposed to the specific method proposed, which appears to overstate the paper’s novelty. Furthermore, the approach warrants a better intuitive explanation. As I understand it, the Fisher Information objective attempts to quantify the sensitivity of model parameters to trajectories expected given some policy. Therefore, maximizing this objective yields a policy that, when executed, yields the maximum additional information about the model parameters.”__
>
> This is correct. As described in Section 3, the inverse Fisher information matrix serves as a fundamental lower bound on parameter estimation error, so exploring to maximize the Fisher information will collect data that will yield the most accurate estimate of the unknown parameters.
>
> Intuitively, the Fisher information quantifies the sensitivity of the trajectory distribution to the unknown parameter. Note that the Fisher information is defined in terms of the gradient of the log-likelihood with respect to the unknown parameter. Thus, trajectory distributions for which the Fisher information is large correspond to trajectory distributions that are very sensitive to the unknown parameters, i.e., trajectory distributions that are significantly more likely under one set of parameters than another. By exploring to maximize the Fisher information, we, therefore, will collect trajectories that are maximally informative about the unknown parameters, since we will observe trajectories that will be significantly more likely under one set of parameters than another.
>
>
> __”Lack of baselining/adequate discussion of other methods that use the Fisher information objective. The statement “As compared to these works, a primary novelty of our approach is the use of a simulator to learn effective exploration policies” seems too strong and overstated given that there are entire fields dedicated to this, and “the application of our method to modern, real-world robotics tasks” is an inadequate claim to novelty.”__
>
> It is certainly the case that using the Fisher information matrix to guide exploration has been proposed before (as our related work makes clear), and, in addition, the use of simulators to aid in real-world robotic learning is also common practice. However, to the best of our knowledge, our work is the first to combine the advantages of using high-fidelity simulators to aid in real-world robotics with active Fisher information-guided exploration for parameter estimation.
>
> In particular, essentially all the works we have cited that rely on the Fisher information matrix to direct exploration (a) do not make use of prior information encoded in simulators to aid in exploration and (b) either do not provide real-world experimental justification for their methods or, if they do, consider only very simple toy problems. In contrast, essentially all the works we are aware of that aim to use simulators to aid in real-world robotic tasks do not perform the type of directed exploration we are proposing to estimate the unknown parameters, but instead use the simulators in a more naive manner, typically by simply transferring a policy that is trained in sim to solves the goal task. Our work can be seen as bringing together these directions—demonstrating that classical works on active parameter estimation can be combined with modern sim2real techniques to solve real-world robotic problems.
>
> If the reviewer is aware of further works that are more relevant to compare to what we have missed in our related work, providing these references would be extremely helpful so we can better situate our work in the existing literature.

---

> > ### Author Response · Authors · 2023-11-18
> >
> > __”In Section 4.2.1, I was expecting to see the standard SysID loss, which is to maximize the likelihood of the data (in this case trajectories) given model parameters. You find the distribution that maximizes likelihood for domain randomization. It seems to me that without some sort of entropy maximization term in the objective, or bootstrap, you would just end up with an MLE objective, whereas it seems like you want to find the Bayesian posterior of models given data. Can you comment on how your objective relates to that of finding a Bayesian posterior?”__
> >
> > We make several remarks on this question. First, our parameter estimation procedure presented in Section 4.2.1 is relatively standard in the literature, see e.g. reference [1] below. Second, we take a frequentist rather than Bayesian perspective in this work: we assume there is a single true parameter that corresponds to real-world physics, and aim to estimate this parameter and use this estimate to perform policy optimization. In such settings, parameter estimators such as the MLE (which our estimator can be seen as a computationally feasible version of) are known to achieve the optimal estimation rates. Furthermore, our parameter estimation procedure is simple, scalable, and works well in practice. We are not aware of any precise connection between our estimation procedure and finding a Bayesian posterior, however.
> >
> > Assuming access to a well-specified prior over parameters, one could also take a Bayesian approach, and replace our estimator with the posterior estimate (e.g. similar to [2]). Such an approach could be incorporated into our framework with some small modifications, but in general, such Bayesian methods are more computationally demanding due to the computational challenges of Bayesian inference. Given this increased computational burden and the effective performance of our estimator in the frequentist setting we consider, we did not consider Bayesian methods in this work.
> >
> > [1] Yevgen Chebotar, Ankur Handa, Viktor Makoviychuk, Miles Macklin, Jan Issac, Nathan Ratliff, Dieter Fox. Closing the Sim-to-Real Loop: Adapting Simulation Randomization with Real World Experience. ICRA, 2019
> >
> > [2] Fabio Ramos, Rafael Carvalhaes Possas, Dieter Fox. BayesSim: adaptive domain randomization via probabilistic inference for robotics simulators. RSS, 2019

---

> > > ### Author Response · Authors · 2023-11-18
> > >
> > > __”For a paper proposing active exploration for the sake of system identification, I wanted to see more discussion of the following: a) regret minimization, i.e. can you prove that your method minimizes regret and achieves the best policy with the fewest interactions with the environment? b) identifiability, i.e. can you say anything about whether all system parameters will be uniquely identified with infinite interactions?”__
> > >
> > > (a) We thank the reviewer for bringing up the regret of our approach, as this is an important feature of our method. It is known that there is a fundamental tradeoff between minimizing standard online regret (i.e. the amount of reward achieved in every online interaction compared to the best possible reward achieved) and exploring to learn the optimal policy: to learn the optimal policy as quickly as possible, one must often explore in a way that incurs large regret (see e.g. Proposition 4.3 of [1] below for a formal discussion of this). Our aim is to explore so as to learn the optimal policy as efficiently as possible and, as such, given the above dichotomy, our approach may not minimize regret (collect a large amount of reward) during exploration. Indeed, in tasks such as rod balancing this is clearly visible: our exploration policy does not aim to pick up the rod (which is required to collect reward), but instead seeks to push it, as this provides more informative data. This is in sharp contrast to the majority of existing sim2real methods we are aware of, which aim to minimize regret during exploration, typically by simply transferring a policy from sim which aims to solve the goal task, and is a key advantage of our approach over such methods.
> > > On whether or not our approach can identify the best policy as quickly as possible, we do not have a formal proof. However, we remark that 1) our approach will collect data to optimally minimize parameter estimation error (see our reply to Reviewer zEL2) and 2) our approach will achieve a significantly faster rate for learning the best policy than existing sim2real approaches which do not explicitly explore as ours does, as is shown by Proposition 4.3 of [1]. There is in fact a subtle and interesting tradeoff between finding the best policy and estimating parameters accurately—certain parameters may be more relevant to finding the best policy than others, and finding the best policy at the optimal rate exploration should be focused particularly on these parameters (this is formalized in the context of linear dynamical systems in [1]). Incorporating this in our method is beyond the scope of this work but is an exciting direction for future work.
> > >
> > > (b) Whether or not the parameter is identifiable depends on the properties of the given environment. It is possible that two different parameters induce identical distributions over trajectories, in which case the true parameter will not be identifiable. We make no assumptions on whether or not the necessary assumptions for identifiability are met. We do remark, however, that in cases where the true parameter is not identifiable, the optimal policy for the downstream task on these simulators would be identical (since for the optimal policies to differ there must be a difference in trajectory distributions). Thus, in cases when the true parameter is not identifiable, identifying it is irrelevant to solving the downstream task, and so will not affect our method’s ability to succeed on this task.
> > >
> > > We will add further discussion on these points to the final version of the paper.
> > >
> > > [1] Wagenmaker, Andrew J., Max Simchowitz, and Kevin Jamieson. Task-optimal exploration in linear dynamical systems. ICML, 2021

---

> > > ### Comment · Reviewer_4bkX · 2023-11-20
> > > **Thanks for your comments**
> > >
> > > Thanks for addressing my questions. I've updated my recommendation to weak accept, but still think this paper could be improved with the following addressed:
> > > 1. In other responses you suggest that other model-based RL algorithms typically leverage fully learned models and not physics based simulators. I don't think this is true, even though a lot of recently popular model-based RL papers might focus on such approaches. Consider the survey of model based RL in Chapter 16 of [1]. No approaches explicitly require the model to be fully learned.
> > > 2. It would be great to baseline against a Bayesian model-based RL like Thompson sampling (16.6 in [1]). I think it would be entirely feasible to construct a Laplace approaximation for the posterior over model parameters given data since you're able to construct the Fisher information matrix. You don't necessarily need to start with a prior over model parameters but certainly could use one.
> > >
> > >
> > > [1] Kochenderfer, Mykel J., Tim A. Wheeler, and Kyle H. Wray. Algorithms for decision making. MIT press, 2022.

---

### Official Review · Reviewer_zEL2 · 2023-10-30

**Soundness:** 3 good
**Presentation:** 3 good
**Contribution:** 3 good
**Rating:** 8
**Confidence:** 4

**Summary:**

This work lays out a framework for robotic manipulation systems to explore and model unknown environments, as well as train a policy to succeed at control tasks within this environment. This generic pipeline for sim to real transfer is called Active Exploration for System Identification, or ASID, and it involves three stages: exploration to gather information about the environment, refinement of this simulation with the data, and training a policy in the learned environment. This approach is shown to be both highly successful and very data efficient, with real work robotics examples shown both in simulation and on real hardware.

**Strengths:**

1. The paper is well written and has a nice flow to it. Organization and structure both help with this as well.
2. The sections on related work and preliminaries do a good job of giving the appropriate context/notation.
3. The tasks chosen to demonstrate this approach were challenging, informative, and speak to the efficacy of the approach.
4. Hardware experiments look convincing.
5. The connections to A-optimal experiment design are insightful and appropriate.

**Weaknesses:**

1. More detail on why the Fisher Information is used vs other methods (observability Grammian, Kalman Filter covariance).
2. The numbers in the heatmaps in Figure 4 are hard to read, maybe block font for the numbers?

**Questions:**

Potential typos:
1. End of section 1 says "signal episode", should this be "single episode"?
2. Section 4.3 says "zero-short", should this be "zero-shot"?

---

> ### Author Response · Authors · 2023-11-18
>
> __”More detail on why the Fisher Information is used vs other methods (observability Grammian, Kalman Filter covariance).”__
>
> As described in Section 3, the inverse Fisher information matrix is a fundamental lower bound on the difficulty of parameter estimation and, in addition, serves as an upper bound when the estimator is, for example, the maximum likelihood estimate. Given this, to minimize parameter estimation error (which is the goal of our exploration procedure), we should aim to explore so as to maximize the Fisher information—doing this will collect data that produces the most accurate estimate of the unknown parameters. Thus, in short, maximizing the Fisher information is the mathematically optimal thing to do if the goal is to minimize estimation error. While criteria other than the Fisher information may be used to direct exploration as the reviewer suggests, they lack this optimality property, and would not necessarily collect data that yields the best possible parameter estimate.
>
> We also remark that using the Fisher information to guide the data collection has a long history in the theory of statistics and experiment design. We refer the reviewer to works such as [1] and [2] below for further discussion of this, as well as formal justification of the argument we have presented above.
>
> [1] Luc Pronzato and Andrej Pazman. Design of experiments in nonlinear models. Lecture notes in statistics, 212(1), 2013.
>
> [2] Friedrich Pukelsheim. Optimal design of experiments. SIAM, 2006.

---

### Official Review · Reviewer_gS1p · 2023-11-04

**Soundness:** 3 good
**Presentation:** 4 excellent
**Contribution:** 2 fair
**Rating:** 5
**Confidence:** 3

**Summary:**

This paper presents a method for active exploration for model based reinforcement learning in the context of robotic manipulation. The paper introduces an exploration policy based on the Fisher information matrix of the parameters of the model. Then, they also include a vision system for scene reconstruction and experimental evaluation based on a robotic manipulator, both in real and simulation environments.

**Strengths:**

The main strength of this paper is the fact that part of the experiments are done in a real manipulator. Also, the pipeline of doing active exploration for model learning (system identification) is fundamental for robotic applications.

**Weaknesses:**

The main weakness for this paper is that this pipeline is very similar to other exploration methods model-based RL. For example:
Shyam P, Jaśkowski W, Gomez F. Model-based active exploration. In International conference on machine learning 2019 May 24 (pp. 5779-5788).

Pathak D, Gandhi D, Gupta A. Self-supervised exploration via disagreement. In International conference on machine learning 2019 May 24 (pp. 5062-5071).

In fact, the pipeline is quite similar to Shyam et al. albeit the metrics and models used are different. However, due to the similarities in the process, those papers should be discussed and, ideally, included in the comparison.

While the experimental section is one of the strengths due to the evaluation in a realistic robotic scenario, the methods should also be evaluated on standard benchmarks for comparison, such as HalfCheetah. The baseline used [Kumar2019] seems very weak (in Fig 4 it does not explore at all). Furthermore, the work of Kumar2019 does not seem to be related to exploration with mutual information as stated in this work.

**Questions:**

-I do not fully understand the reference to REPS as that is a model-free RL method. There is no transition model estimation.
-It seems that the system relies on the assumption that a learned simulator is able to generate accurate trajectories, but that is not the case for out of distribution trajectories. I understand that exploration precisely minimizes that effect, but the probabilistic model should be able to capture the lack of information in out of distribution data. Currently, the only uncertainty comes from the noise if I understand correctly.
-The scene reconstruction part seems to be a part of the specific experiments presented in the paper, but it is unrelated to the exploration pipeline.
-How did you use RANSAC for tracking?
*****
Post discussion update:
If I understood correctly, your method is actually similar to MAX, but instead of using a statistical model as many methods, yours is a physically-informed model, but a parametric model nonetheless. I can see the benefit of using a physically-informed model in a robotics setup. Clearly it is an advantage. However, when evaluating this kind of setups, one has to evaluate the scenario where there are mismodelling errors. For example, most robot models asume rigid-body dynamics, while real life dynamics in high acceleration/forces scenarios suffer from elastic behaviors and therefore are non-Markovian. If the setup is robust, as you said, the policy should be useful (even if suboptimal), but you have to show robustness to mismodeling errors that can make the solution diverge from the actual dynamics.

Also, because you are not learning any policy in section 4.2.1, I wouldn't say that you are using REPS. Instead, if I understood correctly, you are doing supervised learning using natural gradients (which also includes the KL bound). In fact, when you want to do trajectory matching, such as in apprenticeship learning and inverse reinforcement learning, the least-squares loss is problematic, and previous work actually tries to minimize the KL divergence between tau_sim and tau_real directly (see for example: Boularias, Abdeslam, Jens Kober, and Jan Peters. "Relative entropy inverse reinforcement learning." Proceedings of the fourteenth international conference on artificial intelligence and statistics, 2011.)

I agree that the default HalfCheetah does not have variable dynamics, but it can be easily modified (for example, change weight or link length) as has been previously done in other works. For example: https://arxiv.org/pdf/1810.03779.pdf (includes code for some Gym envs).

This is maybe just me being pedantic, but frame/point cloud detection is not tracking. Tracking requires some sequential estimation. Note that this is not a critique on the section: continuous detection might be enough for the experiments. However, as before with the model-based RL it is a suggestion on proper naming conventions to clarify the text.

---

> ### Author Response · Authors · 2023-11-18
>
> __”The main weakness for this paper is that this pipeline is very similar to other exploration methods model-based RL. For example: Shyam et al. Model-based active exploration. ICML, 2019. In fact, the pipeline is quite similar to Shyam et al. albeit the metrics and models used are different. However, due to the similarities in the process, those papers should be discussed and, ideally, included in the comparison.__
>
> __I do not fully understand the reference to REPS as that is a model-free RL method. There is no transition model estimation.
> ”__
>
> We would like to note that ASID is not quite in the same realm as a model-based RL algorithm. Instead, it considers exploring and collecting data so as to identify the parameters of a physics-based *simulator* (such as MuJoCo or PyBullet), rather than an end-to-end model as most model-based RL algorithms would do. In the following, we discuss 1) the conceptual difference of ASID from model-based exploration in terms of explorative behavior and model extrapolation, and 2) provide empirical evidence.
>
> _1) Conceptual difference of ASID from model-based exploration_
>
> ASID fundamentally differs from model-based exploration in the problem setting. Model-based exploration typically leverages fully learned models to guide exploration with the goal of achieving coverage in the state space. The exploration methods proposed by [1] [2] use a learned ensemble of forward models and their disagreement to learn multiple exploration policies that together cover the state space. The collected data is then used to train a downstream task policy. In contrast, our approach relies on the use of an existing physics-based simulator and explores to identify specific unknown parameters in the simulator (e.g. mass, friction coefficients), rather than learning a full model. Our exploration policy searches out state transitions that are most affected by a change in the underlying physics parameters [Appendix Fig. 10c, 10d]. We then roll out a single trajectory in the real world to identify the parameters of the simulator. We use a physics simulator because it extrapolates to unseen states in contrast to a fully learned model [Appendix Fig. 11].
>
> This means we don't learn a dynamics model anywhere in our pipeline and use a model-free black box optimization method (REPS) to identify the physics parameters of our simulator. However, we'd like to note that using a simulator in the process can be seen as a very specific instance of a model-based approach.
>
> _2) Empirical Evidence_
>
> We construct an experiment to differentiate ASID from Model-based active exploration (MAX) [1]. The environment contains four spheres out of which the red sphere’s friction parameters are varied while keeping the parameters of the three blue spheres fixed. As in our previous experiments, the robot uses delta endeffector control (x,y) and is equipped with a peg instead of a gripper [Appendix Fig. 9].
>
> Investigating the exploration behavior, we show that MAX aims to cover the state space through multiple policies throughout training. Since it uses the disagreement between fully learned dynamics models, it gets distracted by novel states induced by the movement of all spheres (red and blue) [Appendix Fig. 10a, 10b]. In contrast, our method based on the Fisher information yields a single policy that seeks out the sphere affected by the changing physics parameters (red) and ignores the irrelevant spheres (blue) even if they lead to novel states [Appendix Fig. 10c, 10d].
>
> With the collected data of both exploration methods, we train a forward dynamics model $s_{t+1} = f_{theta}(s_t,a_t)$. We model $f_{theta}$ as a three-layer MLP and train using the MSE loss until convergence. When evaluated on out-of-distribution trajectories, i.e., trajectories not included in the training data, we find the model to be extremely inaccurate [Appendix Fig. 11]. While the simulator extrapolates to unseen states and correctly predicts the movement of the sphere, the model hallucinates movement even when the endeffector does not interact with it at all [Appendix Fig. 11c, 11d]! These findings make ASID preferable to a purely model-based approach.
>
> [1] Pranav Shyam, Wojciech Jaśkowski, Faustino Gomez. Model-Based Active Exploration. ICML, 2018
>
> [2] Deepak Pathak, Dhiraj Gandhi, Abhinav Gupta. Self-supervised exploration via disagreement. ICML, 2019

---

> ### Author Response · Authors · 2023-11-18
>
> __”While the experimental section is one of the strengths due to the evaluation in a realistic robotic scenario, the methods should also be evaluated on standard benchmarks for comparison, such as HalfCheetah.”__
>
> Standard benchmarks like HalfCheetah don’t include varying physics parameters and, therefore, do not cover the tasks we aim to solve. As shown in the follow-up experiments [Appendix Fig. 10d], our method yields directed exploration toward state transitions affected by the changing physics parameters.
>
> __”Furthermore, the work of Kumar2019 does not seem to be related to exploration with mutual information as stated in this work.”__
>
> The mutual information of a distribution over physics parameters $\Phi$ and corresponding trajectory distribution $\mathrm{T}$ is defined as:
> $I(\Phi|\mathrm{T}) = H\[\Phi] - H[\Phi|\mathrm{T}]$ with $H[\cdot]$ the entropy. With $H[\Phi]$ a fixed quantity, we can approximate $H[\Phi|\mathrm{T}]$ by learning an estimator $q_{\theta}(\phi|\tau)$ from data parameterized by $\theta$ and predicting the physics parameters $\phi$ from a trajectory $\tau$ (c.f. [1]). Minimizing the prediction error of estimator $q$ over $\theta$ (c.f. [2]) then corresponds to maximizing the mutual information.
>
> __”The baseline used [Kumar2019] seems very weak (in Fig 4 it does not explore at all).”__
> The reason [2] performs poorly on our tasks lies in the data used to train the estimator. [2] uses object-centric primitives to collect the data, e.g., the robot always pushing the rod. Every trajectory is, therefore, somewhat informative about the rod's center of mass.
> Since we assume no access to object-centric primitives, we instead use random exploration to collect the initial data and train the estimator. The data now also contains uninformative trajectories from which the parameters cannot be estimated, e.g., because the robot did not interact with the rod. Due to the long horizon, large state and action space, however, the estimator overfits to all trajectories, being equally certain/uncertain about informative and uninformative trajectories. Now, training a policy using the estimator log-likelihood as a reward leads to the policy learning the initial uninformative random exploration behavior used to collect the data.
>
>
> [1] Benjamin Eysenbach, Abhishek Gupta, Julian Ibarz, Sergey Levine. Diversity is All You Need: Learning Skills without a Reward Function. CoRL, 2018
>
> [2] K. Niranjan Kumar, Irfan Essa, Sehoon Ha, C. Karen Liu. Estimating Mass Distribution of Articulated Objects using Non-prehensile Manipulation. 2019

---

> ### Author Response · Authors · 2023-11-18
>
> __”It seems that the system relies on the assumption that a learned simulator is able to generate accurate trajectories, but that is not the case for out of distribution trajectories. I understand that exploration precisely minimizes that effect, but the probabilistic model should be able to capture the lack of information in out of distribution data. Currently, the only uncertainty comes from the noise if I understand correctly. ”__
>
> We assume the simulator perfectly matches the real environment for some existing setting of unknown parameters (cf. Section 3), i.e., the true environment can be modeled by some setting of the simulator. As such, our estimation problem reduces to estimating this unknown parameter, and we do not consider out-of-distribution trajectories since, if we are able to identify the true parameter, the distribution of trajectories generated by the simulator will match the one generated by the real environment. This assumption underlies a variety of applications of sim2real transfer, which assume that the simulator (with appropriately chosen parameters or distributions of parameters) can effectively model the real world, allowing for policy transfer from sim to real [1, 2].
>
> In cases where this does not hold, training a policy in sim is often still a valuable starting point for learning in real [3]. In such settings, learning a targeted distribution of simulator parameters to train a policy for deployment and finetuning the resulting policy in real often leads to improved performance [3, 4]. While such settings violate our assumptions, our exploration procedure would nonetheless generate informative data to learn such targeted distributions of simulator parameters.
>
> [1] Yevgen Chebotar, Ankur Handa, Viktor Makoviychuk, Miles Macklin, Jan Issac, Nathan Ratliff, Dieter Fox. Closing the sim-to-real loop: Adapting simulation randomization with real world experience. ICRA, 2019
>
> [2] Bingjie Tang, Michael A. Lin, Iretiayo Akinola, Ankur Handa, Gaurav S. Sukhatme, Fabio Ramos, Dieter Fox, Yashraj Narang. IndustReal: Transferring Contact-Rich Assembly Tasks from Simulation to Reality. arXiv preprint arXiv:2305.1711, 2023
>
> [3] Laura Smith, J. Chase Kew, Xue Bin Peng, Sehoon Ha, Jie Tan, Sergey Levine. Legged robots that keep on learning: Fine-tuning locomotion policies in the real world. ICRA, 2022
>
> [4] Allen Z. Ren, Hongkai Dai, Benjamin Burchfiel, Anirudha Majumdar. AdaptSim: Task-Driven Simulation Adaptation for Sim-to-Real Transfer. CoRL, 2023
>
>
> __”How did you use RANSAC for tracking?”__
> We use pyRANSAC-3D [1], an open-source implementation for fitting primitive shapes like spheres and cuboids to a pointcloud. The method decomposes a cuboid into 3 plane equations and estimates those by 1) sampling points, 2) fitting a model, and 3) evaluating the model under all available points (c.f. RANSAC). This process is repeated until a threshold criteria or a maximum number of iterations is met. From the plane equations, we can extract the position and orientation of the rod.
>
> [1] https://github.com/leomariga/pyRANSAC-3D/

---

### Official Review · Reviewer_WUcQ · 2023-11-09

**Soundness:** 3 good
**Presentation:** 3 good
**Contribution:** 3 good
**Rating:** 8
**Confidence:** 3

**Summary:**

The paper proposes a system to learn RL policies in simulation which have a high chance of directly trnferring to reality. This is adhieved in a two step process, each step performing RL but with different goals. The goal of the first RL agent is to learn an exploration policy that can collect meaningful simulator calibration data from a single run in the real world. The second RL step learns to achieve the desired goal by learning in a simulator that got calibrated using the once real world run. The main contribution is the first step which uses Fisher information, widely used in system identification, as the cost function of the RL agent.

**Strengths:**

- Treating simulator calibration as system ID is an interesting way to approach things.
- The modifications of the Fisher information to make it suitable for RL training is also nice.
- Paper is well written and easy to understand and follow.
- Outline of questions to answer in the experiments section is a good addition.

**Weaknesses:**

Not per se a weakness of the method but the expectation set by the beginning of the paper. There two aspects that make RL challenging to deploy on real system is safety and sample efficiency. The writing gives the impression that the paper tackles both aspects, when it really tackles the sample efficiency aspect. There is no guarantee that the exploration is safe only that it should be more informative. The proposed approach is interesting as it is and I don't think not dealing with safety aspects is an issue.

Another aspect that does not really fit with the paper is the geometric learning aspect. It does not integrate well with the rest of the paper. The proposed approach is also highly specific and not generally usable. For example, the shape reconstruction is not going to work for complicated objects and will not result in accurate physical simulation outcomes. It is interesting that something like this can be done, but way it is presented and the amount of space available to that aspect makes it hard to fully understand and makes the results sound rather underwhelming.

One aspect that it unclear from the paper is how specific the resulting exploration and task policies are. How generalizable of a policy does the system learn at the end of the day? For example, does the ball pushing policy work only for the specific environment with that breakdown of friction patches and coefficients or is the policy more general and can be used to push balls in a variety of environments? Put differently, do I need to learn a new task policy and calibrate the simulator for every minor varioation of the task description?

The work mentions that it assumes the optimal policy can be found. That is a rather big assumption for RL as finding the optimum is not guaranteed and the other aspect is that often the reward function does not truly represent what we want to optimize for. Does the proposed approach actually need to find the optimum or is a "good enough" policy also acceptable?

Overall the experimental results are nicely presented and show good performance. Two things that could be improved are the discussion of the outcomes. There is little information about failure modes and their explanation, for example. The other part is that Section 5.3. makes sense under the hypothesis that good exploration coverage leads to good RL task performance. Is it possible to show this more directly in that section?

As side comment, maybe using \Pi_{task} for the learned task policy, to mimic \Pi_{exp}, could be a nice way to make it even clearer that there are multiple policies and what their goals are.

**Questions:**

- What is the runtime of the entire system?
- How hard is it to come up with a simulation for the first goal?
- How precise does the simulator have to be?
- There are simplifying assumptions made for the exploration Fisher loss, how limiting are they?
- Equation 4 states that an initial distribution of parameters is assumed, how is this obtained?
- The text states that the system isn't using a differentiable physics engine. If one was used, what would this mean for the method?
- How many parameters can be estimated and what happens when parameters are coupled or jointly multi-modal?

---

> ### Author Response · Authors · 2023-11-18
> **Response to Reviewer WUcQ**
>
> __”One aspect that it unclear from the paper is how specific the resulting exploration and task policies are. How generalizable of a policy does the system learn at the end of the day? For example, does the ball pushing policy work only for the specific environment with that breakdown of friction patches and coefficients or is the policy more general and can be used to push balls in a variety of environments? Put differently, do I need to learn a new task policy and calibrate the simulator for every minor varioation of the task description?”__
>
> The exploration policy generalizes to different physics parameters of the object of interest that behave similarly [Appendix Fig. 12], e.g., mass, friction, and center of mass can all be identified by pushing the object. Furthermore, the policy shows surprising generalization capabilities to sphere locations unseen during training [Appendix Fig. 12]. Furthermore, [Fig. 4] shows that our policy learns to cover the entire space, allowing it to generalize to changes in, e.g., patch locations.
>
> The task policy generalizes to scenarios seen during training. While we assume a zero-shot transfer of our policy, the approach could be extended by domain randomization, i.e., randomizing over object positions and parameters not identified by our system to train a more robust policy.
>
> __”The work mentions that it assumes the optimal policy can be found. That is a rather big assumption for RL as finding the optimum is not guaranteed and the other aspect is that often the reward function does not truly represent what we want to optimize for. Does the proposed approach actually need to find the optimum or is a "good enough" policy also acceptable?”__
>
> We do not rely on finding an optimal exploration strategy. To showcase this, we evaluate different training checkpoints of our policy training in [Appendix Fig. 10d]. The results show that even a “good enough” policy that has not converged to the optimal policy is acceptable for running system identification as most of them perform a reasonable exploration strategy i.e., pushing the sphere that is affected by the parameter variations.
>
>
> __”There are simplifying assumptions made for the exploration Fisher loss, how limiting are they?”__
>
> The simplifying assumptions on the Fisher information loss are not majorly limiting under standard assumptions on the dynamics. In particular, if dynamics take the form $s_{h+1} = f(s_h,a_h)$, as is the case in the setting we consider, then it is often reasonable to assume that there is some small Gaussian perturbation of the state (this is a common assumption, for example, in much of the control theory literature, e.g. canonical settings such as LQG). Given this, our simplified expression for the Fisher Information in Section 4.1 is in fact not a simplification but is exact. In our experiments, we did not observe that using this approximation resulted in a performance loss, even when training a simulator that does not precisely exhibit Gaussian noise.
>
>
> __“The text states that the system isn't using a differentiable physics engine. If one was used, what would this mean for the method?”__
>
> Access to a differentiable physics engine allows one to compute the Fisher information directly. This would make our reward estimate less noisy and stabilize the training of the exploration policy.

---

> > ### Comment · Reviewer_WUcQ · 2023-11-19
> >
> > Thank you for the additional information. Could you also elaborate on the questions outlined in the review?
> >
> > An additional comment based on the replies to the other reviews: It might be beneficial to rethink what aspects of the paper appear in the appendix and which ones do not, as a large number of the questions the reviewers have (and thus future readers will as well) appear to be answered in the appendix. As such it would seem that there is pertinent information that is not part of the main publication.

---

> > > ### Author Response · Authors · 2023-11-20
> > >
> > > Thank you for your suggestions. We will certainly include the additional experiments and results of the discussion as deemed fit in the final manuscript. We have provided answers to your additional questions below, and apologize for missing these in our first reply.
> > >
> > > __What is the runtime of the entire system?__
> > >
> > > Training the exploration policy takes 2-3h on a 12th Gen Intel Core i9-12900K and one NVIDIA GeForce RTX 3090ti. SysID takes 10-20min and downstream task training task 10-20min (CEM and PPO with early stopping upon achieving 100% success on the training env) on a 12th Gen Intel(R) Core(TM) i7-12700KF and one NVIDIA GeForce RTX 3070 Ti. Note that all stages use multiple simulations in parallel and scale with the number of CPUs.
> > >
> > > __How hard is it to come up with a simulation for the first goal?__
> > >
> > > Can you please clarify which environment you’re referring to?
> > >
> > > __How precise does the simulator have to be?__
> > >
> > > We assume that there exists some set of (unknown) simulator parameters such that the behavior of the simulator with these parameters matches that of the real environment (we remark that this or similar assumptions are common in much of the sim2real literature, e.g. [1,2]). However, for parameters other than these ideal parameters, we allow the behavior of the simulator to vary significantly from that of the real world.
> > >
> > > [1] Yevgen Chebotar, Ankur Handa, Viktor Makoviychuk, Miles Macklin, Jan Issac, Nathan Ratliff, Dieter Fox. Closing the sim-to-real loop: Adapting simulation randomization with real world experience. ICRA, 2019
> > >
> > > [2] Bingjie Tang, Michael A. Lin, Iretiayo Akinola, Ankur Handa, Gaurav S. Sukhatme, Fabio Ramos, Dieter Fox, Yashraj Narang. IndustReal: Transferring Contact-Rich Assembly Tasks from Simulation to Reality. arXiv preprint arXiv:2305.1711, 2023
> > >
> > >
> > > __Equation 4 states that an initial distribution of parameters is assumed, how is this obtained?__
> > >
> > > We use the initial parameters settings provided by MuJoCo as a starting point and adjust the mean and standard deviation such that it produces a stable simulation with reasonable-looking/realistic behavior. We find that the initial parameter settings do not matter too much for parameters such as mass, friction, and center of mass, and our method performs well across a range of initial parameter distributions.
> > >
> > > __How many parameters can be estimated and what happens when parameters are coupled or jointly multi-modal?__
> > >
> > > Our exploration procedure naturally handles coupled parameters. Note that the Fisher Information encodes the interaction between parameters, and how jointly varying parameters affect the distribution. Thus, minimizing the inverse trace of the Fisher Information matrix naturally incentivizes exploration that jointly learns coupled parameters.
> > >
> > > Estimating the parameters from a rollout trajectory then depends on the used SysID method and the number of compute budgets available for sampling the simulator. When parameters are coupled or jointly multi-modal black-box optimization methods do not converge to a Dirac delta distribution but instead indicate uncertainty by returning a distribution over parameters. In this case, the downstream task training stage of our method could be augmented by domain randomizing over the returned parameter distribution.

---

### Author Response · Authors · 2023-11-18
**Thank you for your comments!**

We thank all reviewers for their constructive feedback and acknowledging the fundamental importance of reducing model uncertainty, i.e., calibration of a simulator through directed exploration using the Fisher information [WUcQ, gS1p, 4bkX], pointing out the clear presentation [WUcQ, zEL2], and real-world experiments [gS1p, zEL2, 4bkX] of our work.

We address your feedback in the comment section below. You can find an experimental comparison of ASID to model-based exploration and learned dynamics models, as well as an investigation of the generalization capabilities of our exploration policies in [Appendix A.3].

For the final manuscript, we will move the paragraph on the reconstruction pipeline (Section 4.2.2) to the experimental evaluation (Section 5.), add intuition about the Fisher information, and update the abstract. Finally, we want to thank you for the minor comments and additional references that we’ve already added to the rebuttal revision.

---

### Meta-Review · Area_Chair_Rbcp · 2023-12-11

**Metareview:**

This paper presents an approach to address an important problem in real-world RL environments: What data should the agent collect from the real world to improve the policy in a simulator? The presented approach is principled by leveraging fisher information as a measure of system uncertainty. This provides the agent with an objective that can be used to optimize a policy in simulation and then deploy it in the real-world to identify what parameters best represent the environment quickly. The approach is imperfect as it does not consider unmodeled components, but this work represents a good step at improving the training process of RL agents for real-world environments. This paper should be accepted to the conference.

**Justification For Why Not Higher Score:**

N/A

**Justification For Why Not Lower Score:**

Because the technique is not directly applicable to every problem, this paper could be given a spotlight. However, I think it tackles an important problem that is of interest to many in the community. Giving it an oral presentation would stimulate thinking and awareness of this topic that could lead to further advances.

---

### Decision · Program_Chairs · 2024-01-16

Accept (oral)